# Invited Perspective: What Lies Beneath a Changing Arctic?

Jeffrey M. McKenzie[1], Barret L. Kurylyk[2], Michelle A. Walvoord[3], Victor F. Bense[4], Daniel Fortier[5], Christopher Spence[6], and Christophe Grenier[7]

[1]Department of Earth and Planetary Sciences, McGill University, Montreal, H3A 0E8 Canada
[2]Department of Civil and Resource Engineering and Centre for Water Resources Studies, Dalhousie University, Halifax, B3H 4R2, Canada
[3]U.S. Geological Survey, Earth System Processes Division, Denver, Colorado, United States
[4]Department of Environmental Sciences, Wageningen University and Research, Wageningen, The Netherlands
[5]Cold Regions Geomorphology and Geotechnical Laboratory, Department of Geography, Université de Montréal, Montréal, Canada
[6]National Hydrology Research Centre, Environment and Climate Change Canada, Saskatoon, Canada
[7]Laboratoire des Sciences du Climat et de l'Environnement, IPSL/LSCE, CEA-CNRS-UVSQ, Université Paris-Saclay, Gif-sur-Yvette, France

*Correspondence to*: Jeffrey M. McKenzie (jeffrey.mckenzie@mcgill.ca)

**Abstract.** As permafrost thaws in the Arctic, new subsurface pathways open for the transport of groundwater, energy, and solutes. We identify different ways that these subsurface changes are driving observed surface consequences, including the potential for increased contaminant transport, modification to water resources, and enhanced rates of infrastructure (e.g. buildings and roads) damage. Further, as permafrost thaws it allows groundwater to transport carbon, nutrients, and other dissolved constituents from terrestrial to aquatic environments via progressively deeper subsurface flow paths. Cryohydrogeology, the study of groundwater in cold regions, should be included in northern research initiatives to account for this hidden catalyst of environmental and societal change.

## 1 Introduction

Our understanding of congruent hydrologic transformations and climate change in cold regions is derived almost entirely from data collected at or near the land surface from localized field studies or through remote sensing observations (IPCC, 2019). While these studies yield extremely valuable information about shifts in surface water and shallow ground ice distribution, river discharge, and soil moisture (AMAP, 2017; Vaughan et al., 2013), the underpinnings of many of these water-related changes lie beneath the depths of these investigations. Thawing of ancient permafrost is opening and creating new subsurface pathways for groundwater flow (Walvoord and Kurylyk, 2016), thereby altering fluxes and distribution of water, energy, and solutes that can be observed at the Earth's surface. Scientific advances in predicting future climate change require integration of subsurface processes within a broader understanding of change in the Arctic, herein broadly defined to include arctic and

subarctic regions. We argue that groundwater is a catalyst of change in Arctic regions, and we call for a more prominent
inclusion of cryohydrogeology, the study of groundwater in cold regions, in transdisciplinary research initiatives.

## 2 Groundwater - A catalyst of Arctic change

### 2.1 Altered surface hydrology

Present and future groundwater systems in permafrost regions are subject to alteration in response to surface warming because average ground temperature conditions primarily control whether subsurface water is frozen or unfrozen. As Arctic air
temperatures increase and snow patterns change, permafrost is warming and thawing (Biskaborn et al., 2019). Permafrost thaw can increase ground permeability by orders of magnitude (analogous to the stark contrast in permeability of clay versus sand), allowing groundwater to infiltrate and circulate more deeply and across greater lateral distances (Williams and Everdingen, 1973; Walvoord and Kurylyk, 2016). Hydrologic and hydrogeologic regime shifts may occur following the formation of perennially unfrozen zones (taliks) that lie horizontally above the permafrost table and allow for groundwater flow and
transport even during the winter months (Lamontagne-Hallé et al., 2018; Devoie et al., 2019; Walvoord et al., 2019). These changes not only increase the available storage and flux of liquid groundwater, but also enhance the potential for exchange of water between aquifers and surface water bodies (blue lines, Figure 1; Evans et al., 2020; Lemieux et al., 2020) and lead to shifts in vegetation (Christensen et al., 2004).

Historical increases in groundwater discharge during winter months to major rivers (Walvoord and Striegl, 2007; St. Jacques
and Sauchyn, 2009; Duan et al., 2017) and accompanying nutrient and inorganic solute exports are being observed across the pan-Arctic region (Connolly et al., 2020). These data provide compelling evidence that proportionally more precipitation falling on the land surface is being routed through groundwater pathways in response to permafrost thaw. At local to regional scales, increased streamflow has been attributed to thaw-mediated groundwater connections between previously isolated upgradient wetlands and stream networks (Connon et al., 2014). Furthermore, wetter and warmer conditions slow freeze-back,
permitting subsurface pathways for groundwater to persist through the winter. The net result of these changes is increased baseflow and discharge in Northern rivers, particularly during winter. These changes in the subsurface 'plumbing' can explain observed, non-intuitive wetting and drying transformations across the landscape that are manifested at the land surface (Smith et al., 2005; Avis et al., 2011; Lamontagne-Hallé et al., 2018; Pastick et al., 2018). Further, there are implications and feedbacks with vegetation, wildlife habitat, biological productivity, and greenhouse gas emissions (Christensen et al., 2004; McGuire et
al., 2018; Elder et al., 2018).

### 2.2 New transport pathways

Thaw-activated groundwater flow influences the terrestrial to aquatic transfer of nutrients and contaminants in permafrost environments (green and red arrows, Figure 1). Of concern is the fate and transport of globally significant sources of carbon

(Schuur et al., 2015) and mercury (Schuster et al., 2018) stored in permafrost, pathogens (Legendre et al., 2014), and localized anthropogenic contaminants such as organic compounds, heavy metals, and mine tailing runoff. As permafrost thaws, increased groundwater flow and connectivity will become more important for transporting these constituents released from thawing permafrost to aquatic systems (e.g. rivers, lakes) where they are processed or exported to the coastal waters (Tank et al., 2020).

The fate of sequestered organic carbon in thawing permafrost has garnered considerable research attention with focus on decomposition and conversion to greenhouse gases in place, providing a positive feedback to climate change when released to the atmosphere (Schuur et al., 2015). However, large-scale ecosystem models aimed at addressing the strength of the permafrost carbon feedback (Lawrence et al., 2015; Parazoo et al., 2018) typically do not incorporate groundwater geochemical and microbial controls on subsurface processing of carbon, lateral carbon transport, and the potential for storage and burial of permafrost carbon as a mechanism for greenhouse gas attenuation (Neilson et al., 2018; Cochand et al., 2019; Vonk et al., 2019). Though recognized as sources of uncertainty in the net ecosystem carbon balance of permafrost regions (McGuire et al., 2018), these processes remain poorly constrained and thus difficult to adequately represent in ecosystem models due in part to the lack knowledge about groundwater in the Arctic.

## 2.3 Accelerated infrastructure damage

Of paramount concern for Northern societies is the impacts of climate change on transportation infrastructure (e.g. roads, railways, runways), buildings, pipelines, and even trails for access to subsistence resources by indigenous communities (Instanes et al., 2016). As permafrost thaws in ice-rich regions, the ground subsides unevenly leading to thermokarst features and unstable slopes, both of which are destructive to surface structures and incur local and regional costs to society (Figure 1; Instanes, 2016; Hjort et al., 2018). Although sometimes overlooked, changing groundwater flow conditions beneath structures built on permafrost can increase thaw and enhance subsidence rates (Chen et al., 2019). In some cases, erosion and water ponding around subsiding structures requires costly engineering solutions to reroute excess water. Icings (or Aufeis), ice masses due to freezing groundwater seepage, are widely distributed across Northern landscapes (Crites et al., 2020) and can cause flooding and create hazardous conditions on highways, railroads, and airfields. However, little is known regarding the influence of climate-mediated changes to groundwater flow dynamics on the occurrence of icings. Infrastructure designs that typically rely on historical climate information to engineer necessary risk averting measures may be becoming increasingly insufficient to keep pace with rapidly (e.g., decadal timescales) changing climate forcing (Hinzman et al., 2005) and resultant groundwater conditions.. The potential for catastrophic Northern infrastructure failure in specific conditions from a changing groundwater regime presents threats to community security. Although predicting and planning for alterations to thermomechanical conditions that may arise due to changing groundwater conditions is an ongoing challenge, it is of critical importance.

## 2.4 Consequences for Northern water supply

Future climate change will impact Arctic water resources and incur both beneficial and detrimental consequences for water quality and security. Due to the remoteness and extreme cold, water supply and wastewater treatment is very expensive and technologically challenging for Northern communities. Climate change may positively impact Northern water supply in some locations, as previously dormant aquifers will be activated by permafrost thaw, leading to groundwater as a viable alternative or supplemental domestic and municipal water supply (Lemieux et al., 2016). Much work is needed to fully understand the potential for aquifer development in thawing permafrost systems.

Activation of groundwater systems also poses new risks to Arctic water supply. Mobilization of solutes by groundwater can degrade surface and subsurface water resource from pathogens (Legendre et al., 2014) or natural [e.g. mercury (Schuster et al., 2018) or and anthropogenic (e.g. toxic contaminants, landfill leachate) contaminants, raising human and ecosystem health concerns. Also, little is known regarding the impacts of coastal erosion, sea level rise, and saltwater intrusion on Arctic coastal water resources. Some Northern coastal communities are already experiencing saltwater intrusion into surface water reservoirs via surface or subsurface pathways (Johnson, 2018).

## 3. Going below the surface

With projections of rapid and abrupt warming in the Arctic for the foreseeable future, groundwater processes will become increasingly important catalysts of environmental change. Groundwater and lateral chemical transport processes are typically ignored in current Earth System Models (ESMs) used for studying and projecting climate change, even though it is well established that riverine carbon exports, which are strongly influenced by groundwater processes, substantially impact ocean ecosystems and release/burial of carbon in marine environments (Vonk and Gustafsson, 2013). Furthermore, lateral redistribution of surface water and soil moisture across the landscape will impact greenhouse gas exchange in Arctic and Boreal regions, and the exclusion of this process from ESMs limits their ability to foresee and predict cascading effects on the hydrosphere and atmosphere (Fan et al., 2019).

We call for inclusion of *cryohydrogeology* within the larger scope of Arctic climate change research. While there has been limited activity in this field in the past decade, cryohydrogeology research has been conducted in isolation from other Arctic research programs. We propose the following:

- *Northern field programs that address subsurface knowledge gaps are required*. Such programs, while expensive, should be integrated within existing multidisciplinary cryosphere programs. There is a need for improved characterization of Arctic subsurface hydrology to serve as a baseline for future comparison and as input for hydrogeologic and geomechanical models, including ESMs.

- *Groundwater-permafrost feedbacks need to be incorporated into ESMs to improve quantification of regional and global climate projections.* Climate models represent subsurface processes as confined to the vertical dimension with low vertical resolution. Without the incorporation of groundwater flow, lateral transfer of water, solutes (including carbon), and energy that can accelerate the landscape response to surface warming, vertical models may substantially under-represent the rate or magnitude of environmental changes. Recent advances in cryohydrogeological modeling (Grenier et al., 2018; Dagenais et al., 2020; Lamontagne-Hallé et al., 2020) can form the basis for inclusion of lateral processes into Arctic climate change simulations. Incorporation of groundwater dynamics in ecosystem models of permafrost regions will also allow for further exploration of vegetation response (e.g., type, phenology, and productivity) to thaw-mediated water availability. There is also a need for further development of carbon transport via groundwater and conceptual and numerical models that may be incorporated with ESMs.

- *Long-term water management strategies for Arctic indigenous communities, infrastructure, and industry must explicitly consider groundwater* as a potential future water resource, an accelerator of landscape change, and a driver of consequent infrastructure damage and water pollution. As the Arctic thaws, newly mobilized groundwater will directly impact infrastructure sustainability and water security. Water quantity and quality will be influenced by enhanced landscape hydrologic connectivity, altered water residence times, and mobilized contaminants.

Groundwater is a critical component of the Arctic response narrative to climate change and, disregard of hydrogeologic processes by the exciting interdisciplinary, international research programs risks the omission of an important catalyst of change and thereby, limiting understanding (e.g. Sinitsyn et al., 2020). For Arctic development, incorporating hydrogeologic considerations in recommended best practices for design and monitoring of infrastructure overlying permafrost would help Arctic nations and communities actualize sustainable growth and development while balancing economic limitations.

**Author Contribution**

JMM led writing of the manuscript and received support and contributions from all authors.

**Competing interests**

The authors declare no competing interests.

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

325

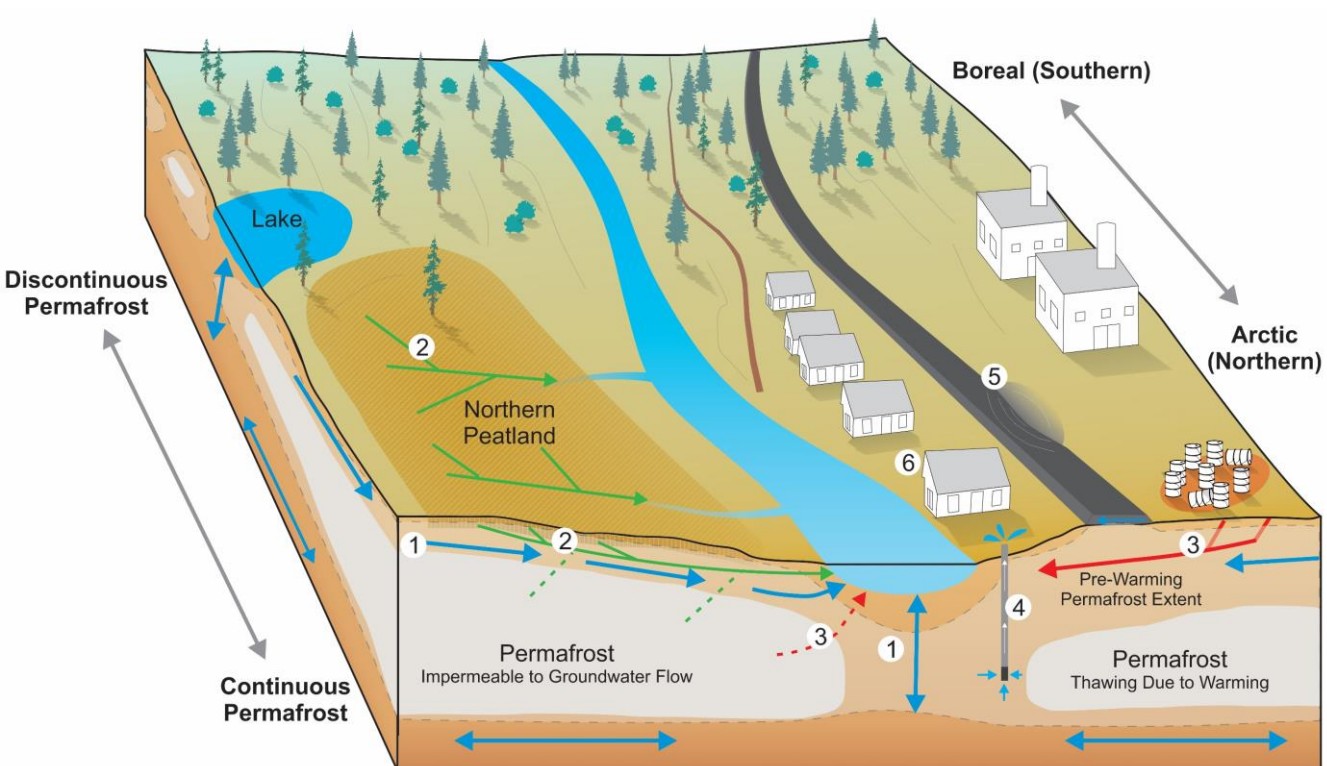

Figure 1: Pathways for groundwater to catalyse environmental change in the Arctic. (1) Arctic warming and permafrost thaw promote increased flux, circulation, and con`nectivity of groundwater above and below permafrost. (2) Groundwater transports carbon and nutrients from terrestrial to aquatic environments via progressively deeper subsurface flow paths with top-down permafrost thaw (green arrows). Permafrost carbon may be mobilized in the aqueous phase upon thaw and transported to inland waters (dashed green lines and arrows). (3) As permafrost thaws, there are opportunities for increased transport of contaminants (e.g. industrial waste, sewage, etc.) due to enhanced groundwater flow (red arrows). Sequestered contaminants, such as pathogens or mercury, are released as permafrost thaws and transported via groundwater flow (dashed red arrow). (4) Water resources will change as permafrost thaws, including increased potential for groundwater development. (5) Groundwater flow

can enhance permafrost thaw rates, leading to land subsidence and destruction of surface infrastructure such as roads or buildings. (6) The incorporation of cryohydrogeology in planning for Northern communities and future economic development would enhance resiliency and fortitude confronting environmental changes.

340