# Peer review of "Invited Perspective: What Lies Beneath a Changing Arctic?"

_The Cryosphere, 2020_

## Referee Comment (RC1) · Sean Carey (Referee) · 18 Jun 2020

The paper by McKenzie and co-authors brings together leaders in cryohydrogeology to provide an invited perspective on how thawing permafrost will influence groundwater in cold regions. They touch on a number of key issues and then present recommendations for future research. Perspective papers are always worthwhile, as it makes the reader reflect on the opinions expressed and more thoughtfully consider issues that may have been ignored by the broader community. They argue that cyrohydrogeology should be included more in transdisciplinary research initiatives. Very fair.

There is little doubt that groundwater is a critical aspect for understanding hydrological and chemical change in permafrost regions as the world warms. The authors state

that it has been limited work in the past decade (line 114), but the issue of permafrost thaw and changes to groundwater has in fact been of interest for many decades now, and while cryohydrogeology is a new term, Van Everdingen, Michel, and others made strong advances in this field over three decades ago. Ultimately, and I agree with the authors, very few people actually study northern groundwater. In contrast to ecological studies in the north, there has not been an 'explosion' of research in hydrogeology (or hydrology for that matter), and in some ways this has deprived earth system modellers and others of a more nuanced understanding of change.

I very much enjoyed reading this article. There have been good review articles on this topic, yet this one is more of an 'agenda setting' document which is nice. That said, and in the spirit of discussion, I have a number of comments that I would like the authors to consider. Perhaps they believe they are out of scope, but this is simply what came to mind after reading the manuscript several times.

+ Is it important to mention that other changes, notably precipitation phase, rate and timing may influence baseflow? This along with the unknown effects of vegetation change? People have long argued that thawing permafrost influences baseflow (of course), but are there other mechanisms that can explain some of this?

+ The authors indicate that earth system models (largely land surface models with biogeochemical processes included) ignore croyhdrogeology. This is largely true! However, cryohydrogeology models largely ignore land surface and biogeochemical processes (particularly with regard to carbon). Surely it is not just the ESM's fault here. Parameterization and incorporation of processes into larger ESM are often incongruent with the granularity that hydrogeological models operate. My comment here is that is this really someone ignoring the issue or not having appropriate tools/guidance on how to address it?

+ Similarly, there are hydrogeological models that ignore freezing/thawing processes that are widely used. This group is well aware of this as they are associated with inter

comparison projects.

+ There is a recent LSM-based paper (Teuful and Sushama 2019) that discusses infrastructure and permafrost thaw. I am curious as to why it is not included on the list? Is it because the LSM largely simulates something that has never been seen and permafrost scientists do not believe the results? This of course reveals my bias for field investigations to advance our understanding of processes. I am often bemused by LSM outputs with sweeping and startling results that are often model artifacts.

+ Would it be helpful to define Arctic? Simply because the issues discussed here are perhaps even more pressing in the subarctic.

+ On Line 85 you state 'rapidly changing groundwater conditions'. Can the authors give an indication of how rapid is rapid? Climate is changing rapidly which immediately affects surface hydrology - can an indication of 'how far behind' the subsurface is be touched upon.

I fully recommend this paper to be published in The Cryosphere. The comments above simply reflect my thoughts upon reading the manuscript and are meant to 'prod' the authors a bit.

Sean Carey

McMaster University

―――――――――――――――――

---

## Referee Comment (RC2) · Anatoly Sinitsyn (Referee) · 16 Aug 2020

The article by McKenzie and co-authors provides philosophical view on the main impacts of groundwater in changing climate in three research areas, namely contaminant transport, modification to water resources, and infrastructure in permafrost conditions. The authors perform "screening" of the effects of groundwater on mentioned above research areas and point out the main consequences. The authors point out the factor of groundwater is overlooked in the analysis of those research areas, and conclude that this needs to be taken into account when setting the research agenda. I agree with the authors.

I may suggest to mention the cascading effects of new knowledge which the authors suggest to develop (in contaminant transport, modification to water resources, and infrastructure) on sustainable development of societies in the Artic. Obviously, the new knowledge build on better consideration of groundwater will help to mitigate the climate change impacts, and will make the Arctic societies more resilient. This would be useful to highlight, especially seeing that this article is aiming not only to scientists, but also to the authorities setting the scientific/research agenda.

The comments I provided are aiming to strengthen the article. I did not find critical point in the article, which I would not accept. Please find my comments in attached PDF.

General scientific comment/wish: Line 26 or 35: I am lacking a sentence or paragraph explaining in more details the reasons behind appearance/activation of the phenomena of groundwater for permafrost. I.e., a link which connects groundwater to the atmospheric processes of higher hierarchy – warming of air temperatures, increase of precipitation, changes in snow patterns (more snow > warmer permafrost) and the permafrost thaw.

Specific scientific comment (see comments to #2.3 in PDF). I have got an impression, that thoughts the authors provide in #2.3 present the impacts of groundwater on infrastructure as something, in a way, new and unexpected/overlooked. It is not always the case. Issues with drainage, needs to reroute excess water, flow of groundwater beneath the structures, eventual floorings along the roads, issue with icings – all such issues usually arise from errors in the initial site investigations/design/constriction/maintenance. These issues may be amplified by the climate change, but the design approaches are normally conservative and able to handle the impact of climate change. These points are reflected in my comments.

In lines 83-85 authors point out that "infrastructure designs that typically rely on historical climate information to engineer necessary risk averting measures are becoming increasingly insufficient to keep pace with rapidly changing groundwater conditions". Would it be the case for the methods utilizing downscaled GCM (see for instance, (Instanes 2016; Incorporating climate warming scenarios in coastal permafrost engineering design – Case studies from Svalbard and northwest Russia)?

I fully recommend this article to be published in The Cryosphere. Hope my comments will be useful for strengthening the article.

Anatoly O. Sinitsyn

Dr., Research Scientist

SINTEF Community (part of SINTEF AS), Group of Rock- and Soil Mechanics, Trondheim, Norway

Please also note the supplement to this comment:
https://tc.copernicus.org/preprints/tc-2020-132/tc-2020-132-RC2-supplement.pdf

**Supplement:**

[revised manuscript text omitted]

* * *
**Commented [AS5]:** Term "coastal ocean" is absent in the arctic coastal studies dealing with engineering and coastal dynamics (coastal erosion). I do not know whether this term is used by the Ecology. Consider using "littoral zone of the ocean".

**Commented [AS6]:** Line 74:

I do not think that the term "overlooked" should be used here. In case if such flow is present on a site then it means that it is a consequence of an error in design/construction/maintenance on the site. Engineering design in permafrost shall assure good drainage around the buildings. Hence, GW is not a factor in foundation design of buildings in permafrost as it is eliminated by the general design approaches. This can be different for design of dams, culverts, even road pavements working partly as dams.

**Commented [AS7]:** Which may point out that the initial design was wrong.

**Commented [AS8]:** Again, this should be revealed during the initial site investigations, then this will not appear as a surprise at the exploitation stage. But sometimes presence (known well before initiation of the project) of icings is simply disregarded in the design.

**Commented [AS9]:** Lines 83-85: Please support with a reference or rephrase by presenting this point as a hypothesis.

Is your suggestion relevant to the methods using downscaled GCM (as for instance (Instanes 2016, Incorporating climate warming scenarios in coastal permafrost engineering design – Case studies from Svalbard and northwest Russia)

**Commented [AS10]:** Lines 85-86: This may be the case for some infrastructures located in the areas with specific site conditions (drainage issues, ground ice content, thickness of unlithified sediment on slope terrain) where groundwater is an important factor for thermal regime of permafrost and/or stability of terrain; or, in broader sense, in the regions with high levels of precipitation now and even higher in the future.

But there are site conditions where precipitation/groundwater will not have such impact (lithified sediment, low ice contend, good drainage).

Hence, such scenario/prediction is not relevant for the whole Northern infrastructure. I suggest you to point out that it is relevant only for some infrastructure/infrastructure under certain conditions.

[...]

**Commented [AS11]:** Fully agree, but for certain problems (water retaining structures, etc.)/certain types of structures (see comments above) /certain types of conditions.

[Figure]

**2.4 Consequences for Northern water supply**

[revised manuscript text omitted]

---

## Author Comment (AC1) · 14 Oct 2020

October 13, 2020

We thank reviewer Dr. Sean Carey for his thoughtful reviews of our invited perspective manuscript, "What Lies Beneath a Changing Arctic?".

Following are Dr. Carey's comments in italics, followed by our response in blue.

*The paper by McKenzie and co-authors brings together leaders in cryohydrogeology to provide an invited perspective on how thawing permafrost will influence groundwater in cold regions. They touch on a number of key issues and then present recommendations for future research. Perspective papers are always worthwhile, as it makes the reader reflect on the opinions expressed and more thoughtfully consider issues that may have been ignored by the broader community. They argue that cyrohydrogeology should be included more in transdisciplinary research initiatives. Very fair.*

*There is little doubt that groundwater is a critical aspect for understanding hydrological and chemical change in permafrost regions as the world warms. The authors state that it has been limited work in the past decade (line 114), but the issue of permafrost thaw and changes to groundwater has in fact been of interest for many decades now, and while cryohydrogeology is a new term, Van Everdingen, Michel, and others made strong advances in this field over three decades ago. Ultimately, and I agree with the authors, very few people actually study northern groundwater. In contrast to ecological studies in the north, there has not been an 'explosion' of research in hydrogeology (or hydrology for that matter), and in some ways this has deprived earth system modellers and others of a more nuanced understanding of change.*

Yes, we agree there has been extensive previous research focused on groundwater in cold regions. In fact, much of the basic theoretical underpinnings of our current understanding are based on research from the 1970s. What is different now is the inclusion of climate change as a strong driver of changing groundwater conditions. Recently, it seems every week there is a new high-profile report, study or news article on the impacts of warming on the Arctic ecology or greenhouse gasses [e.g. 1], but the ecohydrology linkages due to changing surface water and groundwater are usually missing. Hence, part of the motivation for this manuscript. We will add text indicating how our work builds on the historical foundations of Williams, van Everdingen and others [e.g. 2].

*I very much enjoyed reading this article. There have been good review articles on this topic, yet this one is more of an 'agenda setting' document which is nice. That said, and in the spirit of discussion, I have a number of comments that I would like the authors to consider. Perhaps they believe they are out of scope, but this is simply what came to mind after reading the manuscript several times.*

*+ Is it important to mention that other changes, notably precipitation phase, rate and timing may influence baseflow? This along with the unknown effects of vegetation change? People have long argued*
* * *
[1] Harvey, C.: A New Arctic Is Emerging, Thanks to Climate Change, Scientific American, https://www.scientificamerican.com/article/a-new-arctic-is-emerging-thanks-to-climate-change/, 2020.

[2] Williams, J. R. and van Everdingen, R. O.: Groundwater investigations in permafrost regions of North America: a review, in Permafrost: North American Contribution to the Second International Conference, pp. 435–446, National Academy of Sciences, Washington, 1973.

*that thawing permafrost influences baseflow (of course), but are there other mechanisms that can explain some of this?*

Given the strong physical basis for thaw-induced baseflow enhancement, pervasive positive trends in baseflow observed across pan-Arctic permafrost regions, and the contrasting lack of pervasive patterns in precipitation metrics, vegetation change (including wild fires[3]) across pan-Arctic permafrost regions, we contend that permafrost thaw is a primary driver of increased baseflow [e.g. 4]. That said, we appreciate the reviewer's point. The secondary influence of changes in precipitation and vegetation that affect recharge magnitude and seasonality on baseflow in permafrost regions has yet to be well-established and deserves a mention in the revised manuscript.

*+ The authors indicate that earth system models (largely land surface models with biogeochemical processes included) ignore cryhdrogeology. This is largely true! However, cryohydrogeology models largely ignore land surface and biogeochemical processes (particularly with regard to carbon). Surely it is not just the ESM's fault here. Parameterization and incorporation of processes into larger ESM are often incongruent with the granularity that hydrogeological models operate. My comment here is that is this really someone ignoring the issue or not having appropriate tools/guidance on how to address it?*

*+ Similarly, there are hydrogeological models that ignore freezing/thawing processes that are widely used. This group is well aware of this as they are associated with intercomparison projects.*

This is an excellent comment. There is often a gap between the local-scale abiotic cryohydrogeology modeling approaches and the more biochemical ESMs. While we make the case to the ESM community to 'please include groundwater processes!', we will also change the manuscript to note that there is a clear need also for the groundwater community to:

(1) include the transport of solutes, including carbon. There has been some research on this topic, such as Vonk et al. (2019)[5]. Further, there are numerous present initiatives, by some of the co-authors of this paper and others, to include solute transport processes into cryohydrogeologic models.
(2) develop conceptual and numerical methods to incorporate groundwater within ESMs. On the side of catchment scale hydrology and hydrogeology of cold regions, recent advances in cryohydrogeological modeling [e.g. 6,7] can form the basis for inclusion of lateral processes into Arctic climate change simulations or to build spatially distributed reference cases for upscaling projects.

[3] Rey, D. M., Walvoord, M. A., Minsley, B. J., Ebel, B. A., Voss, C. I. and Singha, K.: Wildfire-Initiated Talik Development Exceeds Current Thaw Projections: Observations and Models From Alaska's Continuous Permafrost Zone, Geophys Res Lett, 47(15), doi:10.1029/2020gl087565, 2020.
[4] Qin, J., Ding, Y., Han, T. and Liu, Y.: Identification of the Factors Influencing the Baseflow in the Permafrost Region of the Northeastern Qinghai-Tibet Plateau, Water-sui, 9(9), 666, doi:10.3390/w9090666, 2017.
[5] Vonk, J. E., Tank, S. E. and Walvoord, M. A.: Integrating hydrology and biogeochemistry across frozen landscapes, Nat Commun, 10(1), 5377, doi:10.1038/s41467-019-13361-5, 2019.
[6] Grenier, C., et al.: Groundwater flow and heat transport for systems undergoing freeze-thaw: Intercomparison of numerical simulators for 2D test cases, Adv Water Resour, 114, 196–218, doi:10.1016/j.advwatres.2018.02.001, 2018.
[7] Dagenais, S., Molson, J., Lemieux, J.-M., Fortier, R. and Therrien, R.: Coupled cryo-hydrogeological modelling of permafrost dynamics near Umiujaq (Nunavik, Canada), Hydrogeol J, 1–18, doi:10.1007/s10040-020-02111-3, 2020.

*+ There is a recent LSM-based paper (Teuful and Sushama 2019) that discusses infrastructure and permafrost thaw. I am curious as to why it is not included on the list? Is it because the LSM largely simulates something that has never been seen and permafrost scientists do not believe the results? This of course reveals my bias for field investigations to advance our understanding of processes. I am often bemused by LSM outputs with sweeping and startling results that are often model artifacts.*

We did not cite the publication by Teuful and Sushama[8] as it has led to some disagreement as to the veracity of the results [9,10]. The manuscript uses a LSM to simulate soil drainage and permafrost thaw, and the resulting impact on Arctic infrastructure. Much of the subsequent discussion focused on how subsurface drainage is represented when permafrost thaws and the realism of the results. The paper is an example of the previous comment regarding the need for two-way communication between groundwater focused researchers and the land surface modeling community, and to include lateral water flow and transport.

*+ Would it be helpful to define Arctic? Simply because the issues discussed here are perhaps even more pressing in the subarctic.*

Yes, we will include a definition of our usage of *Arctic* in our revised manuscript. Our definition is broad and is probably best defined as the region north of the southern limit of the discontinuous permafrost zone. Essentially regions that have the presence of perennially frozen ground.

*+ On Line 85 you state 'rapidly changing groundwater conditions'. Can the authors give an indication of how rapid is rapid? Climate is changing rapidly which immediately affects surface hydrology - can an indication of 'how far behind' the subsurface is be touched upon.*

For shallow groundwater systems with little to no data, the best inference of changing groundwater systems is changing patterns of surface water systems (e.g. winter baseflow). Winter baseflow patterns have been observed to be changing over the past few decades, so the changes are happening on a decadal scale [e.g. 11, 12]. It is not clear that changes in shallow groundwater is lagging surface water change. Further, these systems have been in disequilibria for decades, and are continuing to change in response to ongoing climate change. Changes in hydrometeorology are linked to baseflow and vice versa, and the surface water systems may or may not be changing simultaneously.

[8] Teufel, B. and Sushama, L.: Abrupt changes across the Arctic permafrost region endanger northern development, Nat Clim Change, 9(11), 858–862, doi:10.1038/s41558-019-0614-6, 2019.
[9] O'Neill, H. B., Burn, C. R., Allard, M., Arenson, L. U., Bunn, M. I., Connon, R. F., Kokelj, S. A., Kokelj, S. V., LeBlanc, A.-M., Morse, P. D. and Smith, S. L.: Permafrost thaw and northern development, Nat Clim Change, 10(8), 722–723, doi:10.1038/s41558-020-0862-5, 2020.
[10] Teufel, B. and Sushama, L.: Reply to: Permafrost thaw and northern development, Nat Clim Change, 10(8), 724–725, doi:10.1038/s41558-020-0861-6, 2020.
[11] Walvoord, M. A., Voss, C. I., Ebel, B. A. and Minsley, B. J.: Development of perennial thaw zones in boreal hillslopes enhances potential mobilization of permafrost carbon, Environ Res Lett, 14(1), 015003, doi:10.1088/1748-9326/aaf0cc, 2019.
[12] Evans, S. G. and Ge, S.: Contrasting hydrogeologic responses to warming in permafrost and seasonally frozen ground hillslopes, Geophys Res Lett, doi:10.1002/2016gl072009, 2017.

---

## Author Comment (AC2) · 14 Oct 2020

October 13, 2020

We thank reviewer Dr. Anatoly Sinitsyn for his thoughtful review of our invited perspective manuscript, "What Lies Beneath a Changing Arctic?".

Following are Dr. Anatoly's comments in italics, followed by our response in blue.

*The article by McKenzie and co-authors provides philosophical view on the main impacts of groundwater in changing climate in three research areas, namely contaminant transport, modification to water resources, and infrastructure in permafrost conditions. The authors perform "screening" of the effects of groundwater on mentioned above research areas and point out the main consequences. The authors point out the factor of groundwater is overlooked in the analysis of those research areas, and conclude that this needs to be taken into account when setting the research agenda. I agree with the authors.*

*I may suggest to mention the cascading effects of new knowledge which the authors suggest to develop (in contaminant transport, modification to water resources, and infrastructure) on sustainable development of societies in the Artic. Obviously, the new knowledge build on better consideration of groundwater will help to mitigate the climate change impacts, and will make the Arctic societies more resilient. This would be useful to highlight, especially seeing that this article is aiming not only to scientists, but also to the authorities setting the scientific/research agenda.*

*The comments I provided are aiming to strengthen the article. I did not find critical point in the article, which I would not accept. Please find my comments in attached PDF.*

We thank the reviewer for their comments and suggestions. Yes, we agree that incorporation of groundwater management (and how it impacts infrastructure and hydrology) must be part of sustainable development initiatives in the Arctic. We will add text to the revised manuscript to address a broader management framework.

*General scientific comment/wish: Line 26 or 35: I am lacking a sentence or paragraph explaining in more details the reasons behind appearance/activation of the phenomena of groundwater for permafrost. I.e., a link which connects groundwater to the atmospheric processes of higher hierarchy – warming of air temperatures, increase of precipitation, changes in snow patterns (more snow > warmer permafrost) and the permafrost thaw.*

Yes, we agree a clearer description of the linkage between warming and permafrost thaw is required and will include such information, including possibly an additional figure, in addition to the new text from previous comment.

*Specific scientific comment (see comments to #2.3 in PDF). I have got an impression, that thoughts the authors provide in #2.3 present the impacts of groundwater on infrastructure as something, in a way, new and unexpected/overlooked. It is not always the case. Issues with drainage, needs to reroute excess water, flow of groundwater beneath the structures, eventual floorings along the roads, issue with icings – all such issues usually arise from errors in the initial site investigations/design/constriction/maintenance.*

*These issues may be amplified by the climate change, but the design approaches are normally conservative and able to handle the impact of climate change. These points are reflected in my comments.*

While engineering designs are inherently conservative, there are numerous examples, including work from coauthors of this manuscript[1], of situations where warming and permafrost has led to unanticipated engineering challenges. We are amenable in adjusting the text in specific instances in response to the reviewer's comments below.

*In lines 83-85 authors point out that "infrastructure designs that typically rely on historical climate information to engineer necessary risk averting measures are becoming increasingly insufficient to keep pace with rapidly changing groundwater conditions". Would it be the case for the methods utilizing downscaled GCM (see for instance, (Instanes 2016; Incorporating climate warming scenarios in coastal permafrost engineering design – Case studies from Svalbard and northwest Russia)?*

Yes, we agree that downscaled GCMs would be a potentially better method for forecasting future conditions.

**Comments from annotated manuscript.**

*Line 14: General public might understand your phrase as energy goes by "itself", but not transferred by moving water. Just a suggestion, consider to edit: ..."for movements of ground water, which moves energy and solutes".*

Agreed. We will make this change. We would use "transport" instead of "moves".

*Line 15: I would not call it "phenomena", consider using "consequences". To me (PhD in geotechnics), enhances rates of infrastructure damage is not a phenomena, it rather a consequence of errors (design/construction/maintenance) or consequences of applying design philosophies, which appeared to be inadequate (big research question if this can really be the case). But it perhaps OK using phenomena from the formal point of view.*

We agree that "consequences" is a better term and will change text accordingly.

*Line 19: I think that "included" is not fully current. I understand that the authors want to amplify(?) their main point by this word. However, I do not think that groundwater was practically absent in the research initiatives (I do not have a 100% overview of the initiatives), but think there might have been some. Hence, I would suggest pointing out that attention to GW shall be increased in the initiatives. But if there were indeed no initiatives then please keep the phrase as it is.*

Agreed. This comment echoes that of comments from Reviewer 1. We will emphasize that cold regions groundwater research is not new, but that with climate change this research has new and evolving importance. To reiterate, it was not our intention to say that there has been no previous research in this
* * *
[1] Chen, L., Fortier, D., McKenzie, J. M. and Sliger, M.: Impact of Heat Advection on the Thermal Regime of Roads Built on Permafrost, Hydrol Process, doi:10.1002/hyp.13688, 2019.

field, but that as we think about northern change we need to consider featuring groundwater change as a more prominent part of the story.

*Line 29: Here or after the line 35: I am lacking a sentence or paragraph explaining in more details the reasons behind appearance/activation of the phenomena of GW for permafrost. I.e., a link which connects GW to the atmospheric processes of higher hierarchy – warming of air temperatures, increase of precipitation, changes in snow patterns (more snow > warmer permafrost) and the permafrost thaw.*

Yes, as described above we will make this change.

*Line 63: Term "coastal ocean" is absent in the arctic coastal studies dealing with engineering and coastal dynamics (coastal erosion). I do not know whether this term is used by the Ecology. Consider using "littoral zone of the ocean".*

Coastal ocean is very common in oceanography literature, but we will change the term to coastal waters to be broader than the littoral zone.

*Line 74: I do not think that the term "overlooked" should be used here. In case if such flow is present on a site then it means that it is a consequence of an error in design/construction/maintenance on the site. Engineering design in permafrost shall assure good drainage around the buildings. Hence, GW is not a factor in foundation design of buildings in permafrost as it is eliminated by the general design approaches. This can be different for design of dams, culverts, even road pavements working partly as dams.*

Yes, in some cases groundwater is not a concern for design/construction/maintenance. In our experience in northern Canada, groundwater is included but *changing* groundwater regimes are often not included in geotech designs. We will adjust our wording of this sentence to ensure that we do not overstate the case.

*Line 80: Which may point out that the initial design was wrong.*

*Line 80: Again, this should be revealed during the initial site investigations, then this will not appear as a surprise at the exploitation stage. But sometimes presence (known well before initiation of the project) of icings is simply disregarded in the design.*

Yes, we agree that engineering design should account for groundwater systems, including features such as icings. That said, there are many examples we are familiar with where these factors are not properly accounted for. Further, in some settings, features such as icings may change in location from year to year creating further challenges.

*Lines 83-85: Please support with a reference or rephrase by presenting this point as a hypothesis. Is your suggestion relevant to the methods using downscaled GCM (as for instance (Instanes 2016, Incorporating climate warming scenarios in coastal permafrost engineering design – Case studies from Svalbard and northwest Russia)*

We are not in complete agreement that our statement requires a reference. We are simply saying that if historical data were used to project future climate changes without incorporating the dynamics and mechanisms of the changing hydrologic system, the extrapolated result would likely underestimate the change. That said, we will include a refence to support this argument.

*Line 85: Lines 85-86: This may be the case for some infrastructures located in the areas with specific site conditions (drainage issues, ground ice content, thickness of unlithified sediment on slope terrain) where groundwater is an important factor for thermal regime of permafrost and/or stability of terrain; or, in broader sense, in the regions with high levels of precipitation now and even higher in the future. But there are site conditions where precipitation/groundwater will not have such impact (lithified sediment, low ice contend, good drainage). Hence, such scenario/prediction is not relevant for the whole Northern infrastructure. I suggest you to point out that it is relevant only for some infrastructure/infrastructure under certain conditions.*

See response above to Line 74 comment.

*Line 89: Fully agree, but for certain problems (water retaining structures, etc.)/certain types of structures (see comments above) /certain types of conditions.*

Agreed. We do not want to overstate the situation, and we will add a qualifying statement regarding the applicability of this thesis for particular circumstances.

*Line 100: Coastal erosion of permafrost affected coastlines is not always has "thermal" component/driver. For clastic sediment beaches (sandy shores) the "thermal" component of erosion is absent, i.e. all geomorphological work performed by waves only. For cohesive shores (clay, ice-rich sediment) – yes, thermal factor plays a role. Hence, I suggest you avoid using "thermal" here.*

Agreed. We will adjust text accordingly.

*Line 137: This reference might be useful: (Sinitsyn et al., in press), see p. 33. Sinitsyn, A.O., Depina, I., Bekele, Y., Christensen, S., van Oosterhout, D. Development of coastal infrastructure in cold climate. Summary Guideline. SFI SAMCoT report (in press). SINTEF Research 70 Can be requested through ResearchGate: https://www.researchgate.net/publication/338711826_Dev elopment_of_coastal_infrastructure_in_cold_climate_Summ ary_Guideline_SFI_SAMCoT_report_Version_01*

Thanks. We will incorporate this reference.